# The effect of workforce diversity on organizational performance with the mediation role of workplace ethics: Empirical evidence from food and beverage industry

**Abel Tewolde Mehari**[1]*, **Zerihun Ayenew Birbirsa**[2], **Gemechu Nemera Dinber**[2]

**1** Arba Minch University, Arba Minch, South Ethiopia, Ethiopia, **2** Jimma University, Jimma, Oromia, Ethiopia

* abelmgt2000@gmail.com

**Data Availability Statement:** https://dataverse.harvard.edu/dataset.xhtml?persistentId=doi:10.7910/DVN/GEZAJ1.

## Abstract

This study examines the effect of workforce diversity on organizational performance in the Ethiopian food and beverage sector. It incorporates workplace ethics as a mediator to explain in detail the complex relationship between workforce diversity and organizational performance. The study uses a quantitative design, in which data is collected from a random sample of 359 employees from stratified random firms. A Structural Equation Model (SEM) result verified the viability of three of the four proposed hypotheses. Organizational performance is not directly affected by workforce diversity. But it directly changes in response to the change in workplace ethics. Besides, workplace ethics mediate the relationship between workforce diversity and organizational performance. This implies a more advanced relationship exists between diversity and performance, involving effects transitions via workplace ethics, than what this study initially proposed. Thus, an approach to improving firm performance involves reinforcing good workplace ethics practices. Organizations can mitigate the negative impact of workforce diversity on workplace ethics by creating a more inclusive and ethical workplace. This includes providing training on diversity and inclusion and promoting a culture of respect and understanding.

## Introduction

Diversity is a complex and multifaceted concept that is constantly evolving. It encompasses a wide range of individual differences, including gender, ethnicity, sexual orientation, religious beliefs, and physical abilities [1, 2].

It is a critical issue for modern organizations to manage diversity, as it can have a significant impact on workplace ethics, conflict, and organizational performance [3–5]. Managing diversity goes beyond simply recruiting and representing a diverse workforce; it is about creating an inclusive environment where everyone feels valued and respected.

It is not viable to avoid diversity, which could have negative repercussions on the overall operations of an organization. Neither it is viable on the basis of the international legal

**Funding:** The author(s) received ETB45000 that is equivalent to USD789 funding for this work. This is mainly and completely used for field work and related tasks. No other funding is received and all other major costs are covered by the researchers.

**Competing interests:** The authors have declared that no competing interests exist.

frameworks nor in terms of the potential benefits organizations acquire pursuant to successfully managing it. Thus, an appropriate way to approach diversity is by properly managing it.

The issue of diversity management is accompanied by international and national labor policies that indefinitely support Equal Employment Opportunity (EEO) regardless of industry realities. Equal Employment Opportunity (EEO) and diversity are closely intertwined concepts that play a crucial role in creating fair and equitable workplaces. While they are distinct in their specific definitions and applications, they share a common goal: to foster an inclusive environment where individuals are treated equally and valued for their unique contributions. **EEO** focuses on ensuring that all individuals have equal access to employment opportunities regardless of their protected characteristics, such as race, color, religion, sex, national origin, age, disability, or veteran status. It prohibits discrimination in all aspects of employment, including hiring, promotion, compensation, benefits, and training [6]. EEO regulations and enforcement mechanisms are designed to level the playing field and ensure that individuals are not denied opportunities due to factors unrelated to their qualifications or abilities. Diversity, on the other hand, encompasses a broader concept that extends beyond legal compliance and embraces the richness of differences among individuals. It recognizes and values the variety of perspectives, experiences, and backgrounds that people bring to the workplace. Diversity encompasses not only the aforementioned protected characteristics but also other dimensions such as personality, skills, knowledge, and thought styles.

The misalignment between these two emerges from their distinct natures. EEO laws and regulations primarily focus on protected characteristics such as race, gender, and national origin. They may not adequately address other dimensions of diversity, such as personality, skills, and knowledge, which are important for a company's' competitiveness, productivity, and overall organizational performance. Thus, firms wish to comply with the law to prevent sanctions or are merely willing to adopt and apply newly enacted norms [7].

Beyond legal framework and industry reality contradictions, theories contradictorily explain the relationship between diversity management and organizational performance [7]. The theory of heterogeneity illustrates the marginal level of diversity attributes negatively for performance, recommending the suitability of its moderate level [8]. Conversely, social identity theory reflects that individuals have a strong need to belong to groups and that they derive their self-esteem from the positive evaluation of their in-group. When people are exposed to diversity, they may feel a threat to their in-group's identity and status, which can lead to increased prejudice and discrimination. Social categorization theory suggests that people automatically categorize others into social groups based on their perceived characteristics. These categories can lead to stereotypes, which are oversimplified generalizations about a group of people. Stereotypes can be negative or positive, but they can both lead to prejudice and discrimination which ultimately affecting productivity [9].

Besides, empirical findings depict contradictory and inconsistent findings regarding the effect of diversity on performance. Promoting diversity can lead to positive outcomes, such as increased creativity, innovation, and problem-solving [10]. When diverse group members come together, the wide range of perspectives and expertise enhances the quality of debate and improves decision-making; this results in better performance [11–13]. More recent studies show mixed findings, with quite a large number of them implying the merit of diversity for organization performance [2, 14–16], while quite a few of them indicate its adverse effect [17].

Contemporary studies on the relationship between diversity and organizational performance are attempting to incorporate additional variables in to their models by considering mediating and moderating variables to settle the past inconsistent findings. These include perceived organizational justice [18], diversity beliefs, and leadership expertise [19]. However,

additional organizational variables such as workplace ethics also need to be considered, as they can influence the relationship between diversity and performance.

Workplace ethics can be affected by the diversity profile of an organization. For example, negative work ethics can make it difficult to manage diversity effectively, leading to a decrease in productivity, reliability, and accountability, as well as an increase in unprofessional and unhealthy relationships [20].

Either the theoretical explanations or empirical findings signal the contingent nature of the impact of diversity on performance. This indicates the need to account for more organizational variables in contemporary frameworks to activate the effect of workforce diversity on organizational performance.

Therefore, this study aims to create more complete models that include a contemporary concept, workplace ethics, to better illustrate the relationship between diversity and organizational performance.

## Review of related literatures and proposed hypotheses

This section elaborates on the theoretical and empirical relationship among the three variables of the study, *i.e.*, workforce diversity, workplace ethics, and organizational performance. Subsequently, valid hypotheses are proposed.

Workforce diversity refers to the variety of human experiences, backgrounds, and perspectives that individuals bring to the workplace [21]. It encompasses a wide range of differences, including but not limited to race, ethnicity, gender, age, sexual orientation, disability status, personality traits, skills, knowledge, abilities, and experiences [22].

Workplace ethics refers to the principles and standards that guide the behavior and decisions of individuals and organizations in the workplace. It encompasses a range of moral values and norms that promote fairness, honesty, respect, responsibility, and accountability in the professional sphere [11].

Organizational performance is a broad term that encompasses the effectiveness and efficiency of an organization in achieving its stated goals and objectives. It reflects the overall health and well-being of an organization, considering various aspects of its operations, such as financial performance, employee satisfaction, customer satisfaction, and market share [23].

Empirical studies have investigated the relationship between diversity and organizational performance in comprehensive ways, but their findings have been mixed. Some studies have found a positive correlation between diversity and performance, while others have found no correlation or even a negative correlation.

Despite the mixed results, there are a number of theoretical reasons why diversity may benefit organizations. For example, diverse teams may have access to a wider range of perspectives and experiences, which can lead to better problem-solving and decision-making [11]. Additionally, diverse teams may be more creative and innovative, and they may be better able to understand and serve diverse customer markets [23].

Some specific empirical studies have supported the link between diversity and organizational performance. For example, Allen *et al.* [24] found that diversity benefits organizations by allowing them to attract and retain the best talent available; lower costs due to lower turnover and fewer lawsuits; enhanced market understanding and marketing ability; greater creativity and innovations; better problem solving; greater organizational flexibility; better decision-making; and improved overall performance.

Leonard and Devine [25] found that diverse teams are more likely to have access to the breadth of information necessary to solve complex problems. This is because diverse team members bring different perspectives and experiences to the table.

Diversity management is also important for enhancing the link between diversity and organizational performance. When diversity is well-managed, organizations can reap the benefits of diversity while also minimizing the potential challenges. For example, well-managed diversity can help to reduce turnover and absenteeism, attract and retain top talent, and create a more inclusive and supportive work environment [11].

The relationship between these two variables is supported by Human Capital Theory (HCT), which depicts employees as valuable assets whose knowledge, skills, and abilities contribute to organizational success. It states that diversity can enhance the human capital of an organization by bringing together individuals with a wider range of expertise, experiences, and perspectives. This diversity of thought can lead to more creative solutions, improved decision-making, and a more innovative work environment, all of which contribute significantly to organizational performance [26].

Recent studies relating diversity with performance took one dimension of organizational performance, i.e., financial performance neglecting the synergic value of other performance dimensions [27]; emphasize on a specific segment of the organization, i.e., top-level managers [28]; sets limited dimension of diversity as target of study such as age and education [29]. Thus, this study set to recheck the following claim is still persistent in a more inclusive dimension of diversity measurement.

**H1: *Workforce diversity significantly affects organizational performance.***

Besides, the impact of diversity is not limited to organizational performance; it also has a fundamental influence on workplace ethics. Embracing diversity builds stronger teams with better communication and innovation. In contrast, not embracing diversity can have negative ethical consequences. If people feel harassed or discriminated against because of their background or beliefs, this is an ethical issue. It can lead to strain, anxiety, productivity problems, conflict, and even lawsuits [30].

A study of 190 Fortune 500 companies found that diversity management is related to both internal and external corporate ethics [31]. This suggests that companies that recruit and manage a diverse workforce are more likely to have ethical business practices.

Furthermore, signaling theory and social identity theory support the proposed hypothesis (H2). Signaling theory addresses communication by and within organizations [32]. At its core, it is concerned with the roles of the signaler, the signal, and the receiver. Signaling theory offers distinctive insights into how each of these affects the Human Resource Management (HRM) process. It also complements the attributions perspective on HRM. Employees, knowing that the organization promotes diversity, respond with positive ethical behavior, and their commitment increases, subsequently leading to an improvement in organizational performance. The diversity practice is a signal for employees, motivating them to react positively.

Social identity theory (SIT) posits that individuals derive self-esteem and a sense of belonging from their social identities, including their work group or organization. When individuals feel valued and respected for their unique contributions, they are more likely to engage in positive work behaviors such as conscientiousness, dedication, and commitment, leading to enhanced work ethics [9].

Historical studies have sought to investigate the relationship between diversity and workplace ethics and found a valid relationship between those organizational variables. However, their scope is delimited to a single diversity dimension neither containing other diversity dimensions in their model nor utilizing general and perception-scaled measures of diversity [33, 34]. Thus, this study posits to apply a holistic diversity measure, subjectively account for all measurable and immeasurable dimensions, and recheck the validity of the following hypothesis:

*H2*: *Workforce diversity significantly affects workplace ethics.*

Workplace ethics plays a crucial role in shaping an organization's reputation, fostering trust among stakeholders, and ultimately contributing to its success [35]. An ethical work environment cultivates a sense of integrity and transparency, encouraging employees to make sound decisions that align with the organization's values [36]. This, in turn, leads to a more engaged, creative, and trustworthy workforce, positively impacting organizational performance.

Beyond its positive influence on employee behavior, upholding ethical standards also helps mitigate risks and reduce costs associated with unethical conduct [37]. Bribery, corruption, and other forms of misconduct being against ethical standard can erode trust and loyalty, leading to reputational damage and financial losses [38]. A strong ethical foundation, on the other hand, fosters a culture of accountability and compliance, minimizing the likelihood of such damaging behaviors.

Numerous studies have delved into the relationship between workplace ethics and organizational performance; they found that a well-established code of ethics in the service industry positively impacted performance and compliance [39].They emphasized the need for organizations to proactively address ethical concerns and implement stringent measures against violations. They also examined the link between ethical standards and organizational productivity. While they found a positive correlation between ethical standards and productivity, they also observed a negative relationship between integrity and discipline and increased productivity. This suggests that the abstract nature of these virtues may require longer-term observation to fully assess their impact [40].

A study also investigated the role of ethical leadership in corporate social responsibility (CSR) and its subsequent influence on organizational performance. Their findings revealed a favorable impact of ethical leadership on CSR, which in turn enhanced organizational performance. They further concluded that CSR partially mediates the relationship between ethical leadership and organizational performance [41].

Agbim [42] explored the connections between ethical leadership, corporate governance, performance, and social responsibility. Their study demonstrated a strong positive impact of ethical leadership on all four dimensions. They advocated for integrating corporate governance, performance, and social responsibility under the leadership of an ethical leader to build a strong organization.

Abidin *et al.* [43] investigated the relationship between ethical commitment and financial performance. They found a positive correlation between the two, suggesting that a strong commitment to ethical principles contributes to financial success.

Painter-Morland and Dobie [44] focused on the impact of ethical business practices on the long-term sustainability of small and medium-sized enterprises (SMEs). They concluded that SMEs should prioritize embedding ethical principles into their operations to ensure long-term viability and productivity.

Kehinde [45] investigated the effects of ethical behavior on organizational performance. Their findings indicated a positive correlation between ethical behavior and organizational performance, highlighting the positive impact of ethical conduct on organizational success.

Vieira [46] examined the relationship between a firm's ethical performance and its financial profitability. It revealed a significant positive correlation between the two, suggesting that ethical practices contribute to financial gains. These studies underscore the multifaceted benefits of workplace ethics, ranging from fostering trust and enhancing performance to mitigating risks and promoting long-term sustainability. By embracing ethical principles and cultivating an ethical work environment, organizations can reap significant rewards, contributing to their overall success and fostering a positive impact on society as a whole.

Furthermore, instrumental stakeholder theory and Resource based view supports and logically illustrate the relationship between workplace ethics and organizational performance, supporting the proposed hypothesis, *H3*.

Instrumental stakeholder theory emphasizes the instrumental value of ethical conduct in achieving organizational goals. According to this theory, ethical behavior can lead to enhanced financial performance by fostering positive relationships with stakeholders, such as customers, employees, and investors [47]. When an organization demonstrates ethical behavior, it attracts and retains customers who value ethical practices Ethical behavior also contributes to increased employee engagement and productivity, as employees feel more valued and respected in an ethical work environment. Additionally, ethical practices can attract and retain investors who prioritize ethical conduct in their investments. These positive stakeholder relationships, in turn, contribute to increased sales, reduced employee turnover, and a lower cost of capital, ultimately boosting the organization's financial standing.

The resource-based view of the firm posits that ethical behavior serves as a valuable resource that provides a competitive advantage. Organizations known for their ethical practices attract and retain high-quality employees, customers, and partners [48]. This leads to enhanced innovation, increased productivity [48], and a stronger market position [49], contributing to overall organizational success.

Therefore, evidence suggests that workplace ethics is an important factor affecting organizational performance. Organizations can benefit from building a strong ethical culture, reducing ethical risks, and adhering to high ethical standards.

Studies relating ethics with performance were conducted on Fortune 500 listed firms using archival data disregarding the relevance of primary data [31]. Besides studies emphasize a specific segment of organization staff(leadership positions) disregarding the role of promoting ethical values among ordinary employees for firm performance [50, 51]. Thus, this study is aims to engender the neglected segment of the organization and to test the following hypothesis.

***H3: Workplace ethics significantly affects organizational performance.***

Empirical evidence concerning the relationship between diversity and firm performance is mixed and context-dependent, being mediated by work ethics [52].

Fair and equitable recruitment practices which gives value to diversity can signal to employees that the organization cares about diversity and is committed to everyone, which may lead employees to act more ethically and contribute to the organization's internal ethics and subsequently to productivity [53, 54]. Further, studies have shown that employees want to work for companies that care about them, and they respond to such caring with positive attitudes and behaviors [54, 55]. Firms perceived as having positive reputations for ethics were found to attract more competitive and diverse workforce talent [56, 57] and achieve higher commitment from those already employed [58] in part due to greater trust, job satisfaction, and value congruence.

To clarify, when organizations have a culture of diversity and inclusion, employees are more likely to be motivated and engaged ethically, which in turn leads to better organizational performance.

There is a dearth of studies investigating the mediation role of ethics in organizational studies. A more related study conducted recently found that workplace ethics reinforces the relationship between implementing HR practices (Recruitment and selection) and organizational performance. However, still, this study still fails to recall the relevance of diversity and inclusiveness to affect workplace ethics [59]. Thus, this study establishes and tests the following hypothesis to fill this empirical discrepancy.

**H4**: *Workplace ethics mediates the effect of workforce diversity on organizational performance*

Moreover, moral capital theory supports the proposed research model in Fig 1 and all the proposed hypotheses (H1-H4). Moral capital theory states that performance is higher in an organization or group with strong moral values such as fairness, integrity, and reliability. This theory can be quite instructive when considering the advantages of diversity and moral behavior.

Diversity can enhance moral capital in a number of ways. Diverse teams bring together people with a wider range of experiences and perspectives, which fosters creativity and innovation [23]. This diversity of viewpoints leads to the questioning of assumptions and improves decision-making [60]. In addition, a culture of inclusive and open communication encourages respect and trust between team members [61]. This broad sense of trust and respect is very helpful in building a strong foundation of moral capital.

Additionally, developing moral capital requires ethical action. Employee trust and security are increased in the workplace when they observe and feel that leadership makes just and moral decisions [62]. Robust moral capital deters unethical behavior such as taking shortcuts or hurting others, resulting in a more efficient and general healthier work atmosphere [63] In addition to its internal advantages, ethical behavior enhances the organization's reputation and helps it draw and keep top personnel while fostering stronger bonds with partners and consumers [64]

Moral capital has a beneficial effect on productivity inside a business. Productivity increases when moral capital is strong because it fosters cooperation and teamwork [65] As innovation and taking risks are necessary for progress, a culture of openness and trust that is supported by moral capital is also favorable to these activities [66]. Strong moral capital also helps with conflict resolution and problem solving, which reduces interruptions and boosts productivity [65]. These benefits are cumulative. All of these things lead to a high-performing firm, and moral capital essentially lays the groundwork for trust, collaboration, and creativity, all of which support an organization that operates at peak performance.

The structural relationship among workforce diversity, workplace ethics, and organizational performance is presented by Fig 1.

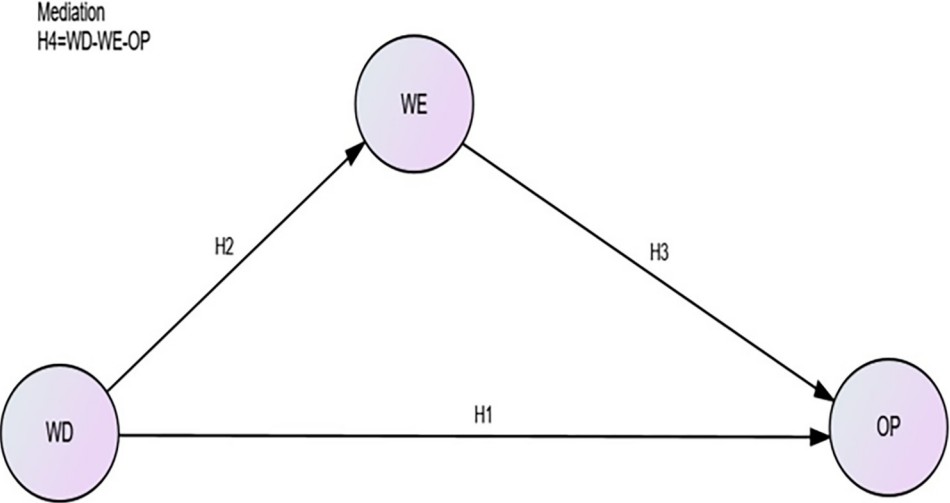

**Fig 1. Proposed structural model.**

## Methodology

### Data collection tools and the instrument

A standardized questionnaire is used to collect the required data from randomly selected employees of firms in the food and beverage industry in Ethiopia. The.xlxs and.sav formats of the data, along with ancillary documentation, the questionnaire, and the data dictionary, are accessible at Harvard Dataverse [67]. The questionnaire was developed by adopting nine latent constructs along with their corresponding measurement items from three discreet past studies [14, 68, 69].

Diversity is measured using two reflective latent constructs, which later merge into a second-order latent construct. These dimensions show higher reliability and validity based on the three criteria of evaluation (Eigenvalue Cronbach's alpha, and percentage variance explained).

Eigenvalues are used in factor analysis to determine the number of factors that underlie a set of items [70]. Each factor represents a common dimension of the items, and the eigenvalue for each factor indicates the amount of variance that the factor explains [71]. A higher eigenvalue indicates that the factor explains more of the variance and that it is therefore a more important factor [72].

Cronbach's alpha is a measure of internal consistency, which is how closely related the items on a scale are to each other [73]. A high Cronbach's alpha value (usually above 0.7) indicates that the items are measuring the same thing and that the scale is reliable [74]. In other words, if you were to administer the scale multiple times to the same group of people, you would get similar results each time [75].

The percentage variance explained is the percentage of the total variance in the items that is explained by the factors [70]. A high percentage variance explained (usually above 50%) indicates that the factors are doing a good job of explaining the variance in the items [7]. In other words, the factors are capturing most of the variability in the items [8].

The Eigen value, Cronbach's alpha, and percentage variance explained for Promoting Diversity (PD) are 4.91, 0.55, and 0.88, respectively. Likewise, these three values for Retaining Diversity (RD) are 3.72, 0.53, and 0.84, respectively.

Workplace ethics is measured using an instrument adopted from the work of Park and Hill [69].They developed the shortest form, the Occupational Work Ethic Inventory Short Form (OWEI-SF), which has 12 items and adequate GFIs. Another study conducted by Hill and Fouts [76] suggested that each factor of the OWEI showed acceptable reliability coefficients for internal consistency with 373 responses: interpersonal skills (r = 0.90), initiative (r = 0.88), and being dependable (r = 0.78). The findings here in this study confirmed the reliability of the construct IS with a Cronbach's alpha of 0.80, an Eigen value of 2.52, and a percentage of variance of 0.63. The construct IN has a Cronbach's alpha of 0.88, an Eigen value of 3.34, and a percentage of variance of 0.67. DE has a Cronbach's alpha of 0.84, an Eigen value of 2.29, and a percentage of variance of 0.76.

The study adopted a reflective assessment model to subjectively estimate organizational performance [14].The Eigen value, percentage of variance explained, and Cronbach's alpha measures are 4.54, 0.75, and 0.93, respectively. Likewise, Blouch and Azeem [18] also used this particular measurement and confirmed its suitability with the factor analysis result of an Eigen value of 3.51, a percentage variance of 0.58, and a Cronbach's alpha of 0.53.This particular study also confirmed the instrument's suitability with an Eigen value of 3.81, a percentage variance of 0.64, and a Cronbach's alpha of 0.89.

## Context, sample size and sampling technique

This study focuses on the manufacturing industry, specifically the food and beverage segment. This segment employs more people than any other sub-sector in Ethiopia and is the largest sub-sector in terms of employment, followed by rubber and plastic goods and textile firms [77].

Furthermore, compared to other sub-sectors in the industry, food and beverage has a considerable number of establishments. As of the year 2016/17, the total number of firms in the sub-sector is 947. These firms are scattered among 21 public and 926 private. The private food and beverage companies hired 52522 workers, and the public firms employed 11913 workers [77]. Thus, the subsector hires a total of 64435 employees. This serves as the total target population of the study (N).

A minimum sample threshold is computed using the following Cochran formula [78].

$$n_0 = \frac{Z^2(p)(q)}{e^2} \tag{1}$$

$$n = \frac{n_0}{1 + \frac{(n_o - 1)}{N}} \tag{2}$$

Where,

$n_0$-Sample size for infinite population

n-Sample size for finite population with finite error correction

N-Population size

Z-Confidence level z score

P-Proportion based on population characteristics

e-Margin of error

$$\frac{1.96^2(0.5)(0.5)}{0.05^2} = 385 \tag{3}$$

$$n = \frac{385}{1 + \frac{(385 - 1)}{64435}} \tag{4}$$

$$n_e = 382 \; employees$$

*Firm level sample size is computed as follows:*

$$n_0 = \frac{1.96^2(0.5)(0.5)}{0.05^2} = 385 \tag{5}$$

$$n = \frac{385}{1 + \frac{(385 - 1)}{757}} \tag{6}$$

$$n_f = 254 \; firms$$

Therefore, with the final computation, the minimum valid sample size is set to be 382 employees of food and beverage firms, or 254 firms. Further, considering the complex nature of the proposed model and the possible non-response rates, the actual sample size is raised by 5 percent, and it is finally set at 400 employees.

Subsequently, a total of 254 firms were randomly selected from a complete list of food and beverage firms in Ethiopia. Using an Excel random number generator, firms in each strata (A–M) were randomly selected based on the proportion indicated in Appendix Table 1 in S1 Appendix. For all strata, the required number of firms was obtained as designed in Appendix Table 1 in S1 Appendix, except for strata C, E, and H. In these exceptional strata, the actual number of available firms was less than that of the required firms. Hence, all these firms were completely taken as sample firms.

The strata is an industrial classification obtained from a 2016 report by the Central Statistics Authority. Brief description of all thirteen strata presented in Appendix Table 1 in S1 Appendix is presented below:

> **Strata A**: *Covers the farm-to-table journey of meat, fruits, and vegetables, including raising/ growing, processing, and preservation.Strata B: Makes edible oils and fats from plants and animals.* **Strata C**: *Turns milk into cheese, yogurt, ice cream, and other dairy products.* **Strata D**: *Mills grains (wheat, rice) into flour, cereal, and animal feed.* **Strata E**: *Makes custom animal feed for poultry, cattle, etc.* **Strata F**: *Bakes bread, pastries, cakes, cookies, etc.* **Strata G**: *Makes sugar and candy from sugarcane/beets.* **Strata H**: *Produces pasta (macaroni, spaghetti, etc.) from flour and water.* **Strata I**: *Catches all other food production (snacks, frozen meals, etc.).Strata J: Makes spirits (whiskey, vodka, etc.) through distilling and aging.* **Strata K**: *Makes wine from grapes (fermentation, aging).* **Strata L**: *Brews beer and malt beverages.* **Strata M**: *Makes soft drinks, juices, and bottled water (filtered, flavored, carbonated).*

The study randomly selected a sample of 400 employees from food and beverage firms in Ethiopia (details in Appendix Table 1 in S1 Appendix). Employees' responses (359 usable out of 400) serve as the data to analyze performance, ethics, and diversity within these firms.

## Data analyses tools and process

The aim of the study is to test and verify the validity of the hypotheses set (H1-H4) in the Ethiopian food and beverage industry. One of the relationships has an indirect and extended path; thus, SEM is taken as an appropriate tool of analysis.

The commonly recommended and also used statistical tool in this study is SPSS-AMOS.It is a powerful software package for SEM analysis with large sample sizes and reflective measurement models [79, 80].Likewise, entirely the measurement models in this study have merely reflective items, with all regression arrows pointing outward from the latent variables to the measurement items.

The assumption of normality is one of the preconditions for CFA/SEM. Non-normal data in SEM can lead to misleading p-values and underestimated standard errors, requiring robust methods or data transformations [81].Thus, the normality of the data was checked using AMOS's built-in tools. Univariate and multivariate normality tests of skewness and kurtosis were performed. Acceptable values for skewness fall between -3 and +3, and for kurtosis between -10 and +10 [82]. However, the univariate skewness and kurtosis results for all 34 items were not within these ranges.

Efforts were made to remove outliers to improve normality, but this did not bring any change to the distribution of the data. Therefore, an alternative estimation method, B-S bootstrapping, was used to address the potential bias caused by the non-normality of the data. B-S Maximum likelihood estimation (B-MLE) is a robust estimation method that is less sensitive to violations of normality than other estimation methods, such as maximum likelihood (ML) estimation. This makes it a recommended alternative estimation method to minimize bias

when the data is not normally distributed [83–85]. However, it is necessary to verify that the data is suitable for bootstrapping before using it. This has been checked and reported in Tables 2 and 4.

B-S Bootstrapping is a resampling procedure that can be used to reduce estimation bias in non-normal data. The default number of resamples in AMOS is 200, but the commonly accepted threshold is 1000. The minimum recommended bootstrapping sample in AMOS is 1000 [86]. However, a larger sample size may be necessary to obtain accurate estimates for certain statistics. For example, a sample size of 2000 or more may be necessary to obtain accurate estimates of confidence intervals for the median of a population. Thus, all CFA first-order (Fig 2), CFA second-order (Fig 3), the SEM model (Fig 4) are estimated with 1000 bootstrapped resamples.

The bootstrap results indicated that the fit of the resampled data was not significantly different from the fit of the actual data, confirming the appropriateness of bootstrapping. Also, the sample CMIN was within the implied bootstrapped range. Similarly, the bootstrap results for the second-order CFA, and the SEM model were also valid, with p-values of 0.175, and 0.175, respectively. The sample CMIN for all three models were within the ranges of the bootstrap distributions.

Common Method Bias (CMB) is a serious challenge when using Likert scales or perception scales. It is a type of measurement error that can occur when the same respondents answer all of the items in a questionnaire. CMB can influence the significance, magnitude, and direction of coefficients [87].

A statistical test (Herman single-factor) reported by Table 1 confirmed low risk of bias from the survey design (CMB) in this study (20% variance explained, well below the 50% threshold).The analysis ran into identification problems with the statistical models (CFA & SEM). To fix this and get reliable results, researchers adjusted the models.Therefore, parameters of 9 items in the first order CFA are set to 1; parameters of 12 items are fixed to 1 in the second order CFA, and the SEM model. Subsequently, estimated results are computed and presented in figures and tabular formats.

## Analyses and results

The study used a multi-step statistical approach to ensure reliable findings. First, exploratory factor analysis (EFA) cleaned the data and identified underlying structures. Then, confirmatory factor analysis (CFA) validated the measurement and confirmed the concepts were accurately measured. Finally, structural equation modeling (SEM) tested the relationships between the variables, providing a comprehensive understanding of how factors influence performance, ethics, and diversity.

## Exploratory factor analysis and common method bias tests

The rotated component matrix reported in Appendix Table 2 in S1 Appendix shows the items which have good quality for latent construct measures. The estimation is based on suppression of items with factor loadings less than 0.5 with varimax rotations and fixing the total number of factors extracted to 9.

An entire of 34 Likert scale items were set to measure 6 latent constructs(factors).However, only 31 of these items met a recommended and acceptable factor loadings of above 0.5.Thus 4items were left blank, unreported and removed from subsequent estimation because these items could compromise the data quality and leads to estimation biases.

The analysis result in Appendix Table 2 in S1 Appendix confirms 2 items (pd8 & pd9) were deleted from the latent construct, PD. These items were insignificantly loading either to the

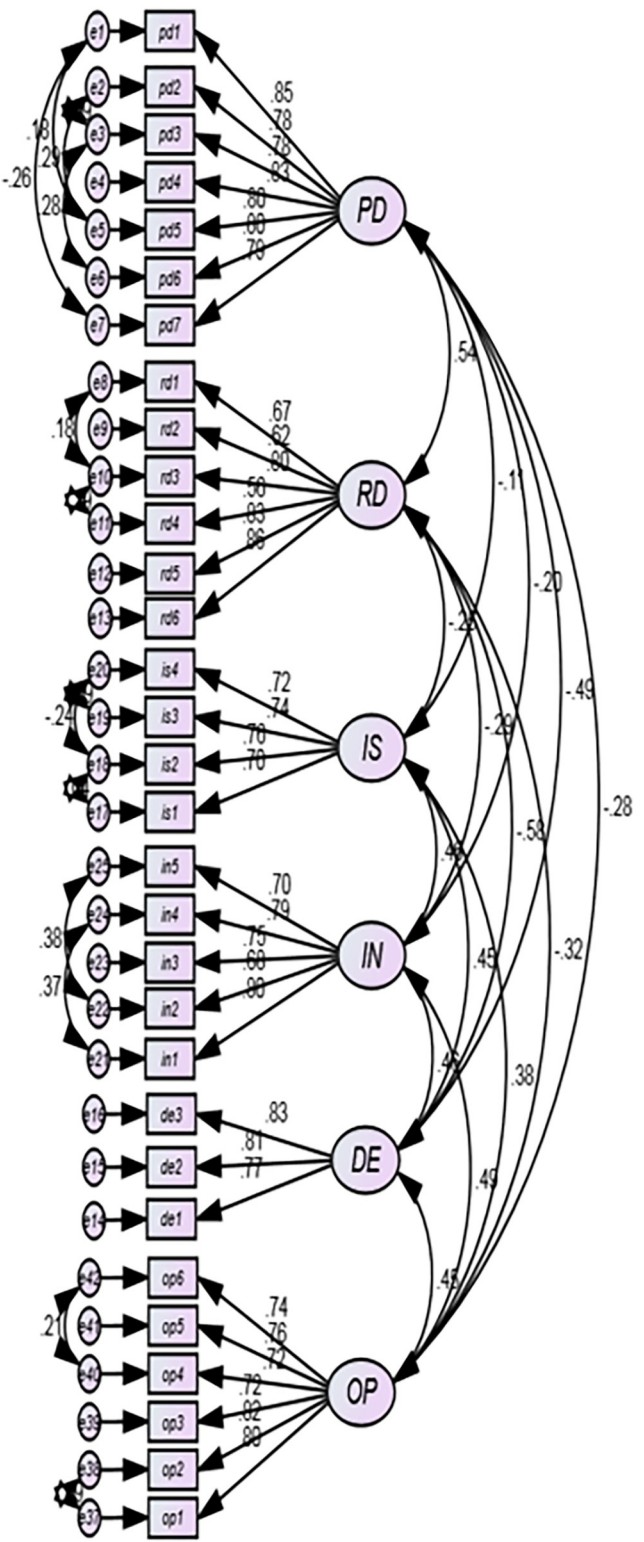

**Fig 2. First order CFA of the proposed final model.**

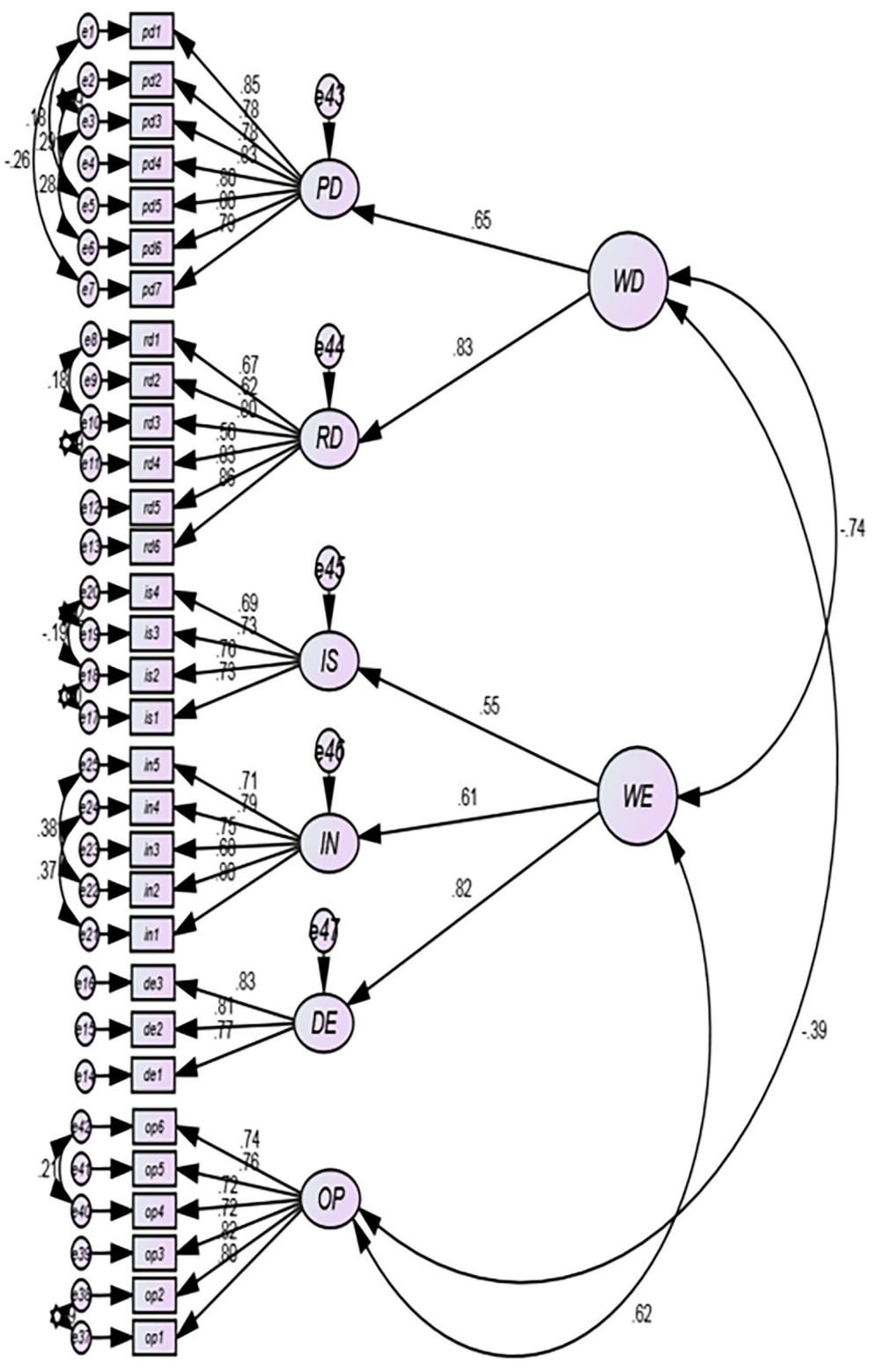

**Fig 3. Second order CFA of the proposed final model.**

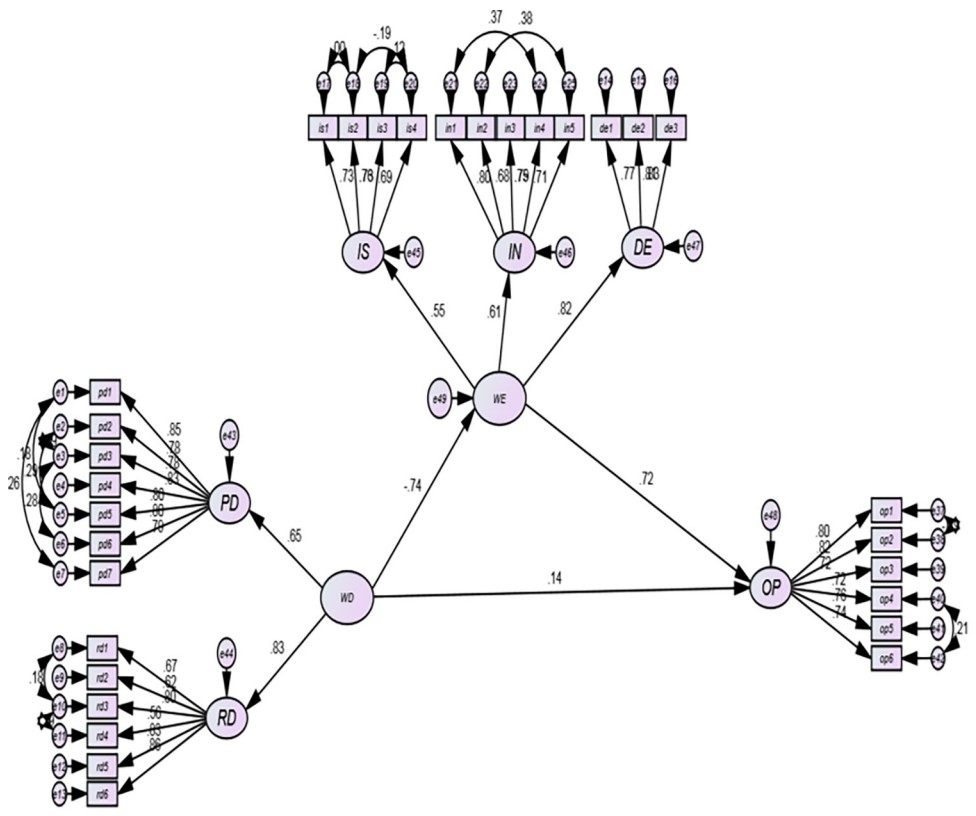

**Fig 4. Structural model (final model).**

construct or any other latent constructs. Likewise, a single item (rd7) was deleted from a latent construct, RD, ultimately having 6 items.

All the items adopted from Park & Hill [69] load sufficiently on the three latent constructs. The 4 items on IN, 5 items on IS, and 3 items on DE loads properly and adequately.

The entire 6 items properly loads on the latent construct Organizational Performance (OP). The lowest factor loading of items in this construct is 0.72, *i.e.*, far beyond suggested threshold.

Common Method Bias (CMB) test result is reported in Table 1. The total percentage variance explained is 27.57 percent when all the items are forced to load on a single construct. An instrument having a CMB, the percentage variance should be above 0.50. Thus, there is no problem of CMB pertaining to the instrument.

**Table 1. Herman's single factor test for CMB.**

| Component | Initial Eigenvalues | | | Extraction Sums of Squared Loadings | | |
|---|---|---|---|---|---|---|
| | Total | % of Variance | Cumulative % | Total | % of Variance | Cumulative % |
| 1 | 9.373 | 27.567 | 27.567 | 9.373 | 27.567 | 27.567 |
| 2 | 4.172 | 12.271 | 39.838 | | | |
| 3 | 2.440 | 7.178 | 47.016 | | | |

**Source**: Own Survey, 2023

## Confirmatory factor analysis and the measurement models

The entire study has three measurement models, as depicted in Fig 2 under the LoC model. These are used to assess different realities in the Ethiopian food and beverage industry. The first measurement model is workforce diversity (WD). It has two sub dimensions, *i.e.*, Perceived Diversity (PD) and Retaining Diversity (RD). PD is meant to evaluate how firms in the industry are conducive and capable of attracting diverse workforces and what management techniques they implement to achieve this. RD is supposed to measure the firms' efforts and ways of retaining or maintaining diversity.

The second measurement model, Workplace Ethics (WE), comprises three sub dimensions. The sub dimensions are reflective by nature, *i.e.*, Interpersonal Skill (IS), Initiative (IN), and Dependability (DE).

Items in IS are meant to measure the soft skills of an employee in his or her relationship with peers, subordinates, and supervisors. One of the items in this sub dimension is *"Do you use words like excuse me, sorry, or please in your communication"*. The latent construct, IN, is composed of items that indicate the employee's degree to lead the initiative for innovation, task completion, and industriousness. One of the items in this set is *"Do you do more than is required or expected of you?"* The third sub dimension in this measurement model, *i.e.*, *DE*, is all about how an employee is obedient to rules, regulations, and orders. One of the items in this category is *"Do you do more than is required or expected of you?*

The fourth and last measurement model depicted as part of Fig 2 is set to measure organizational performance. It is entirely a subjective assessment. Unlike the previous three models, this model is a lower-order CFA. All the items directly assess and reflectively measure an ultimate variable, organizational performance. The items are financial, operational, and market-based indicators.

Generally, the model in Figs 3 and 4 are Reflective-Reflective Measurement Model, known as the Second-Order Construct Type I, one of the models most frequently applied in SEM among researchers [88].The measurement models for WD, and WE, have major latent and sub-latent dimensions in addition to the outer measurement items. Distinctly, organizational performance is considered as a first-order measurement model.

Higher-order CFAs are advised to be evaluated in such a way that the lower-order construct (LoC) CFA is initially assessed and a similar procedure is applied sequentially to subsequent higher-order CFAs. Specifically, this actual study is a second-order CFA; thus, the first-order (LoC) CFA was evaluated first, followed by the second-order (HoC) CFA.

Scholarly documents illustrates four ways of estimating higher order models. These include repeated approach [89], two step approach [90], mixed two step approach, and the PLS Components Regression Approach [91].

All the SEM estimations presented here in this section are based on repeated approach [89]. The repeated indicators approach is the first and the most popular approach: the indicators of the LOCs are used as the manifest variables of the HOC [92]. Conversely, the two step approach, as described by Cataldo [91], "consists of two phases: first, the latent variable scores of the LOCs are computed without the HOC [93]; then, the regression analysis is performed using the computed scores as indicators of the HOCs. The implementation is not performed through a single regression; this implies that any Second-Order Construct, investigated in stage two, is not taken into account when estimating the LV scores in stage one"

**Assessment of lower order constructs (reliability, validity, and model fit).** The LoC (first-order) CFA model presented by Fig 2 encompasses 34 items and 6 latent constructs. Initially, 34 items were considered, but an Exploratory Factor Analysis (EFA) was used to check and screen the data quality. Subsequently, three items possibly having problems with cross-

loading and small factor loadings were removed, considering factor loading below 0.5 (the EFA Rotated Component Matrix is presented in Appendix Table 2 in S1 Appendix) as a criterion of omission.

The latent construct PD, which was initially proposed to have nine items has seven items, of which two are deleted. The other latent construct, RD, deleted 1 item and became a 6-item construct. Finally, the first-order CFA model presented in Fig 2 is viable for B-S bootstrap estimation at the 5 percent level of significance.

The minimum Standardized Regression Weight (SRW) in the LoC CFA presented in Fig 2 is 0.62, and the maximum ranges up to 0.85. Therefore, LoC's items are valid based on the advice that items in a CFA with SRWs above 0.5 [85]or above 4 [94] could be kept in the model.

The model fit index reported in Table 2 is a goodness of fit index for the LoCs CFA model. The commonly used goodness of fit indexes to evaluate a CFA are Comparative Fit Index (CFI), Tusker-Lewis Index (TLI), Root Means Square Approximation (RMSEA), and Standardized Root Mean Squared Residual (SRMR) [95, 96]. However, here almost all fit indexes are reported for the merit of readers.

A CFA model's fitness is admissible when the CFI and TLI are greater than or equal to 0.90 [97] and the RMSEA and SRMR are less than 0.05 [98]. Studies also illustrate that CFI and TLI values of 0.85 could also be acceptable, and RMSEA and SRMR values up to 0.1 are acceptable [99].The Chi square ($\chi2$)/degrees of freedom (CMIN/df) value for a model is acceptable if it is less than 2, and it is also tolerable even if it reaches 5 [100]. The $P$ value of this particular indicator should also be insignificant [101]. In accordance with these threshold values, the LoC CFA model fitness is admissible with CMIN/df = 1.92, CFI = 0.93, TLI = 0.92, RMSEA = 0.04, and SRMR = 0.06, as illustrated in Table 2.

The LoC CFA is free from validity and reliability problems. Values in Table 3 illustrate the validity and reliability of the LoCs CFA model. Composite Reliability (CR) and AVE for all constructs are above the suggested threshold 0.7 and 0.5 [98, 102].

Discriminant validity is confirmed using the Fornell and Lacker [103] test result presented in Table 3. According to the Fornell-Larcker criterion [103], the discriminant validity of a latent construct is established when the square root of its Average Variance Extracted (AVE) exceeds all its correlations with other latent constructs. This implies a latent construct should explain more variance of its indicators than is explained by other latent constructs. For example, the diagonal value for the latent construct DE is 0.803, and the correlation of all other constructs corresponding to this latent construct (In = 0.461, IS = 0.454) is below 0.803. Thus, all

**Table 2. Fit test statistics for the LoCs of measurement model.**

| Measures | Suggested Values | Measurement model |
|---|---|---|
| B-S P value | >.05 | .301 |
| CMIN/df | <3.00 | 1.54 |
| NFI | >0.90 | 0.91 |
| RFI | >0.90 | 0.90 |
| CFI | >0.90 | 0.97 |
| IFI | >0.90 | 0.97 |
| TLI | >0.90 | 0.96 |
| RMSEA | <0.08 | 0.04 |
| SRMR | <0.08 | 0.04 |

**Source**: Own Survey,2023

**Table 3. Construct reliability, convergent and divergent validity for LoCs.**

|    | CR | AVE | MSV | MaxR(H) | DE | PD | RD | IN | OP | IS |
|----|------|------|------|------|------|------|------|------|------|------|
| DE | 0.845 | 0.645 | 0.342 | 0.848 | **0.803** |  |  |  |  | 0.454*** |
| PD | 0.927 | 0.645 | 0.292 | 0.929 | -0.488*** | **0.803** |  |  |  | -0.114† |
| RD | 0.872 | 0.537 | 0.342 | 0.898 | -0.585*** | 0.540*** | **0.732** |  |  | -0.252*** |
| IN | 0.861 | 0.555 | 0.238 | 0.867 | 0.461*** | -0.200** | -0.290*** | **0.745** |  | 0.463*** |
| OP | 0.891 | 0.578 | 0.238 | 0.895 | 0.447*** | -0.284*** | -0.316*** | 0.488*** | **0.761** | 0.384*** |
| IS | 0.821 | 0.534 | 0.214 | 0.822 |  |  |  |  |  | **0.731** |

Significance of Correlations:† p < 0.100

* p < 0.050

** p < 0.010

*** p < 0.001

**Source:** Own Survey,2023

the latent constructs do not have any concern for divergent validity. Moreover, a cross-loading report table which is not presented here in this manuscript, indicates no indicator cross-loads beyond the construct with which it is theoretically associated. According to Gefen and Straub [104], discriminant validity is shown when each measurement item correlates weakly with other constructs except for the ones to which it is theoretically associated.

**Assessment of higher order constructs (reliability, validity, and model fit).** The HoC Confirmatory Factor Analyses (CFA) presented in Fig 3 encompass all 31 observed items along with their corresponding lower-order latent constructs. The first-order CFA presented in Fig 2 was taken as a baseline to estimate the second-order CFA.

Uniquely, three additional latent constructs were added in Fig 3 in addition to the first-order CFA presented in Fig 2. The newly added latent constructs are assumed to be Higher-order Constructs (HoCs).These constructs are designed as a reflective latent on LoCs. The CFA model in Fig 3 shows these newly added HoCs as WD, meant to measure Workforce Diversity, and WE for Workplace Ethics.

The lowest and highest SRW in the second-order CFA model are 0.56 and 0.87, respectively. Specific to the HoC reflective link with LoCs, the lowest and highest SRWs are 0.55 and 0.83, respectively. Therefore, the HoC CFA model meets the minimum suggested SRW threshold of 0.40 [94].

**Table 4. Goodness-of-fit statistics for the LoCs and HoCs of measurement model.**

| *Measures* | *Suggested Values* | *Measurement model* |
|----|----|----|
| B-S P value | >0.05 | 0.175 |
| $X^2$/df | <3.00 | 1.63 |
| RFI | >0.90 | 0.90 |
| NFI | >0.90 | 0.90 |
| CFI | >0.90 | 0.96 |
| IFI | >0.90 | 0.96 |
| TLI | >0.90 | 0.95 |
| RMSEA | <0.08 | 0.04 |
| SRMR | <0.08 | 0.06 |

**Source**: Own Survey,2023

**Table 5. Construct reliability, convergent and divergent validity for HoCs.**

|  | CR | AVE | MSV | MaxR(H) | OP | WD | WE |
|---|---|---|---|---|---|---|---|
| OP | 0.891 | 0.578 | 0.386 | 0.895 | **0.761** |  | 0.621*** |
| WD | 0.713 | 0.558 | 0.543 | 0.750 | -0.395*** | **0.747** | -0.737*** |
| WE | 0.708 | 0.454 | 0.543 | 0.758 |  |  | **0.674** |

Significance of Correlations

† p < 0.100

* p < 0.050

** p < 0.010

*** p < 0.001

**Source**: Own Survey,2023

**Source:** Own Survey,2023

The HoC CFA model fit index is quite good, meeting the suggested threshold values. In Table 4, the fit index values are reported as CMIN/DF = 2.07, CFI = 0.92, TLI = 0.91, RMSEA = 0.05, *and SRMR = 0.07*. This entire fit index illustrates that the hypothesized HoC model adequately fits with the observed covariance matrix.

Reliability and validity concerns are not detected in the HoC CFA model. The CR and AVE values for the majority of the constructs are above 0.67 and 0.50, confirming the convergent validity of the higher-order latent constructs. The exception is for WE (AVE = 0.454), its convergent validity is confirmed by the relaxed suggestion of using the CR of the construct instead of AVE [103]. The AVE of this particular construct is below 0.5, *i.e.*, 0.454, but its CR is above 0.6. Thus, based on Fornell and Lacker's [103] suggestion, WE fulfills the convergent validity requirement.

The discriminant validity of the HoCs is indicated by the Fornell and Lacker sections in Table 5. The diagonal values for each HoC are higher than the correlation of the construct with other HoCs except for WD and WE.

Generally, the regular model fit index, B-S bootstrapping, convergent validity, and discriminant validity for both LoC CFA and HoC CFA are acceptable and valid. Thus, conducting a structural model analysis is permissible and viable.

## Structural model assessment /hypotheses testing

The structural model presented in Fig 4 illustrates the relationship between the designed core variables of interest: Workforce Diversity (WD), Workplace Ethics (WE), and Organizational Performance (OP).

The SEM model in Fig 4 originates from the second-order CFA model presented in Fig 3. The B-S bootstrapping is valid and fits with a p value of 0.175. The procedure advises a higher p value, confirming the fitness of the actual sample covariance matrix and implied data covariance matrix [105].

The structural model in Fig 4 is the hypothesized model without control variables. WD serves as the sole independent variable; WE as a mediators; and OP as a dependent variable. All the fit index reported in Table 6 for this structural model are within the acceptable ranges were corresponding suggested values are included.

The hypotheses test results presented in Table 7 are for the four proposed direct effects and a simple mediation. As illustrated, three of them are direct hypotheses (H1-H3), the last is simple mediation (H4).

**Table 6. Fit test statistics for the structural model.**

| Measures | Suggested Values | Measurement model |
|---|---|---|
| B-S P value | >0.05 | 0.175 |
| $X^2$/df | <3.00 | 1.63 |
| NFI | >0.90 | 0.90 |
| RFI | >0.90 | 0.89 |
| CFI | >0.90 | 0.96 |
| IFI | >0.90 | 0.96 |
| TLI | >0.90 | 0.95 |
| RMSEA | <0.08 | 0.04 |
| SRMR | <0.08 | 0.05 |

**Source**: Own Survey,2023

## Discussions

Employee diversity has been increasingly recognized as an important factor for organizational performance. Businesses that embrace diversity tend to experience positive outcomes, including increased creativity, better decision-making, and improved profitability [106]. Promoting diversity is especially relevant in multiracial countries, as it can lead to a number of benefits, including improved market performance, innovative performance, and stakeholder performance [107–109]. Thus, organizations need to create a diverse and inclusive work environment that allows employees to feel valued and appreciated, irrespective of their backgrounds. Leaders and managers should make diversity a priority and strive to create a work culture that values diversity, fosters inclusion, and promotes equality [110].

Workforce diversity in organizations is believed to increase productivity and organizational performance. Turi *et al*. [19] used four objective measures of diversity dimensions (age, gender, education, and ethnicity) and confirmed that only one of these dimensions (age) significantly affects organizational performance. Likewise, Song, Yoon, and Kang [111], illustrate gender diversity having a positive and significant effect on firm performance while age diversity has an insignificant effect on firm performance.

Contradictory or mixed facts were also reported pertaining to the effect of workforce diversity on organizational performance. Kundu and Mor [14] conducted a study on employees of IT firms in India and found a positive effect in one of the four diversity indicators, while three of them were not significant enough to show the intended effect. Particularly in the food and beverage industries in Ethiopia where this study targets, workforce diversity does not pose any direct effect on the organizational performance (**H1:WD→OP**) of firms at any statistical significance level. The beta coefficient of the statistical result($\beta = 0.14, P = 0.272$) in Table 7 shows

**Table 7. Hypothesis test result and decision without controls.**

| Hypothesis | Standardized Estimate(B) | P-value | Decision | Type |
|---|---|---|---|---|
| H$_1$:WD→OP | .14 | .272 | Unsupported | Direct |
| H2:WD→WE | -.74 | .002 | *Supported* | Direct |
| H$_3$:WE→OP | .72 | .001 | *Supported* | Direct |
| H4:WD→WE→OP | -.53 | .000 | *Supported* | *Indirect* |

Squared Correlation($R^2$): OP = 0.40,WE = 0.54
**Source**: Own Survey,2023

the positive effect of promoting and retaining diversity on organizational performance, but this is not a valid and statistically significant effect supported by the existing data. One of the possible reasons for not detecting the direct effect of workforce diversity on organizational performance is its complicated relationships and implications with other organizational variables. This has been shown by the valid effect of the transiting variable, *i.e.*, WE, in the relationship between the two variables of the study. This implies the power of workplace ethics in breaking the status quo of the relationship between diversity and performance.

While firms promoted and managed diversity, they spent effort recruiting new employees to maintain a consistent structure of employee inventory in terms of various diversity dimensions (gender, age, and education). This effort brings new employees with different experiences and cultures to the firms. Subsequently, existing workplace ethics will be adversely affected [31]. Workforce diversity significantly affected the important proposed mediating variable. This likely paves the way for its possible indirect effect on organizational performance.

Promoting workforce diversity leads employees to doubt employees who have peculiar characteristics. This could adversely affect workplace ethics, and employees could show unethical behavior in response to the situation [112].The finding of this study shows employees in the food and beverage industry show unethical behavior(reduction in workplace ethics behavior) at the workplace in response to the prevalence of high diversity at the workplace ($\beta = -0.74, P = 0.002$).Thus, the proposed hypothesis (**$H_2$:WD→WE**) is supported at 1 percent level of significance. Research has shown that diverse teams may be more likely to engage in unethical behavior, such as groupthink and conformity [113]. This is because diverse teams may have a more difficult time reaching consensus, and team members may be more likely to go along with the majority opinion, even if they disagree with it [114].

The change in workplace ethics could affect organizational performance [115, 116]. This has been proposed as a separate hypothesis (**$H_3$:WE→OP**).The result in Table 7 ($\beta = 0.72$, $P = 0.001$) shows significant effect of workplace ethics on organizational performance at 1 percent statistical significance. Work ethics has been shown to have a positive impact on performance, as employees with strong work ethics are more likely to be productive, efficient, and effective in their jobs [117]. Ethics is the central axis that helps organizations, corporations, and enterprises to achieve favorable financial outcomes, job balance, and well-being by balancing risk. On the other hand, a lack of ethical principles application might have a detrimental impact on individuals, resulting in harm, severe penalties, and even bankruptcy.

Diversity, by itself or by its physical nature, cannot have an impact on organizational performance. For this dimension to have either a fruitful or destructive role, it has to be perceived by employees. Subsequently, employees feel comfort or discomfort in their routine activities. One can consider diversity as a stimulus and the reaction to follow as the behavioral change of employees working in a diverse environment. The behavioral change could have a constructive or destructive impact on organizational performance [118].

There are a number of ways in which diversity can be a stimulus for behavioral change in employees. One way is by challenging the status quo. When employees are exposed to different perspectives and ways of thinking, it can force them to question their own assumptions and biases. This can lead to a more open-minded and inclusive approach to work, which can be beneficial for both individuals and organizations.

Another way in which diversity can be a stimulus for behavioral change is by providing new opportunities for learning and growth. When employees from different backgrounds come together, they can share their knowledge and experiences with each other. This can help employees develop new skills and competencies, which can be beneficial for both their careers and the organization.

Recent studies have applied a more advanced model (moderation/mediation) to see the relationship between diversity and performance but fail to provide concrete and holistic direction regarding how firms should encounter diversity and a policy to be pursued regarding it. Khassawneh and Mohammad [119] aim to investigate the effect of income diversity and race diversity on organizational performance via training in the UAE hospitality sector. The study illustrated that income-level diversity has been negatively moderated and race diversity has been positively moderated in training, which in turn has a positive mediating impact on performance. However, it says nothing about the effect of diversity on performance mainly emphasizes the attenuation and reinforcement of employee training in the relationship between the two organizational variables.

Organizational-level studies are complicated. No direct effect does not mean no total effect or indirect effect [120]. The absence of a valid direct effect does not signify the total absence of its effect on organizational performance. There are scenarios where effects could be dormant and wait until other factors activate them. Likewise, WD could have hidden or indirect effects on organizational performance via other relevant factors. Thus, a mediation hypothesis is set (H4: WD→WE→OP)

Flourishing workplace ethics implies tight and formal work conditions, which are convenient only for skilled and capable employees. Such an environment could be seen as a threat to non-skilled people and those who prefer informal over formal environments. Thus, these employees might get into conflict with those who adhere to and act according to the norm of workplace ethics. Synonymously, the sentiment of us and them will develop, which finally leads to conflict in the organization.

An intense ethical can might cause ethical conflict. Ethical conflict is "a situation in which there is a disagreement between two or more individuals or groups about the ethical implications of a decision or action [121]." Ethical conflict can arise in any organization, but it is more likely to occur in organizations with high ethical standards. This is because employees in these organizations are more likely to be sensitive to ethical issues and to have strong opinions about what is right and wrong [121]. Employees who perceived their organization to have a high ethical culture were also more likely to report experiencing ethical conflict. This suggests that ethical environments can create a breeding ground for conflict, even if the conflict is ultimately rooted in positive intentions. Ethical conflict can lead to moral disengagement, which is a psychological process that allows people to justify unethical behavior. Moral disengagement can have a negative impact on workplace performance, including increased absenteeism, turnover, and decreased productivity. Ethical conflict is a two-edged sword, it is linked to poor team dynamics, such as lower group satisfaction, viability, cohesiveness, psychological safety; increased negative emotions; and perceived goal difficulties [122].

Promoting and retaining workforce diversity as an organizational diversity management policy was not able to contribute directly to the advancement of organizational performance. However, this does not mean that diversity does not have any other effect on it. This is reinforced by the mediation analysis result. The possible mediation is valid through workplace ethics. The result in Table 7 indicates support for the hypothesis (H4: WD→WE→OP) workplace ethics mediates the effect of workforce diversity on organizational performance at the 1 percent statistical significance level ($\beta = -0.53$, $P = 0.00$). Thus, workplace ethics fully mediates the effect of workforce diversity on organizational performance, given that workforce diversity does not have a direct statistically significant link with organizational performance. Thus, workforce diversity reduces organizational performance via workplace ethics.

## Conclusion and implications

Workforce diversity does not have a direct effect on organizational performance. However, workplace ethics directly reinforce firm performance; a flourishing ethical environment motivates employees to be productive. This has been vividly supported by the result in Table 7, where the effect of workplace ethics on organizational performance is significant at the 1 percent level of significance.

A diverse environment disturbs the ethical environment. This has been confirmed by the empirical results of this study. Similarly, a less diverse environment does not compromise norms and ethical values in the industry. This is reinforced by the analysis result depicted in Table 7, in which the relationship between workforce diversity and organizational performance is valid at the 1 percent level of significance.

Moreover, the study found that workplace ethics fully mediates the relationship between workforce diversity and organizational performance at the 1 percent level of significance, as illustrated in Table 7. This means that the impact of workforce diversity on organizational performance is entirely due to its impact on workplace ethics. Full mediation implies the total effect of WD on OP is just the indirect effect, which passes via workplace ethics. This finding reinforces the proposed overarching theory, i.e., moral capital theory. However, failure to have a significant relationship between diversity and workplace ethics that could be considered against the overarching theory, implies the presence of hidden and omitted organizational variables that need to be considered broadly and inclusively in redefining and improving the existing theory.

Practically, the finding implies organizations effort of promoting diversity in their premises will not be fruitful unless it is reinforced or supported by good workplace ethics because merelypromoting diversity negatively affects workplace ethics. Thus, organizations can mitigate the negative effect of workforce diversity on workplace ethics by creating a more inclusive and ethical workplace. This includes providing training on diversity and inclusion, promoting a culture of respect and understanding, creating policies and procedures that support diversity and inclusion, and addressing any ethical concerns frequently raised by employees.

Organizations should also monitor the impact of workforce diversity on workplace ethics and implement interventions to mitigate any negative impacts. This could include conducting surveys of employees to assess their perceptions of workplace ethics and developing and implementing training programs to address any identified problems.

Creating a culture that values and rewards ethical behavior is also important. This could include developing and implementing a code of ethics, providing training on ethics, and recognizing and rewarding employees who demonstrate ethical behavior.

Generally, while workforce diversity in the food and beverage industry is important, simply increasing the number of employees from different backgrounds might not directly benefit the organization. In fact, the study suggests it could have a negative impact on workplace ethics. To address this, HR professionals need to focus on fostering inclusion alongside diversity. This means implementing training programs that promote understanding and respect, developing clear policies against discrimination, and establishing channels for employees to voice concerns and report ethical breaches. Regularly monitoring employee perceptions and mitigating any negative effects of diversity through targeted interventions is crucial. Finally, creating a strong ethical culture through a code of ethics, training, and recognition programs will ensure a diverse workforce thrives while maintaining high ethical standards. This comprehensive approach will ultimately lead to a more productive and successful organization.

## Limitations and future directions

This study emphasizes and it is only limited toto Ethiopian food and beverage industry, manufacturing segments of the economy. The issue of diversity could have been distinct if the study targets the service sector. The working environment of the service sector is quite different from manufacturing sector were the later involves additional stakeholder, especially user, on spot in addition to ordinary employees and managers. This fact complicates the relationship between diversity, ethics, and performance. Thus, the generalizability of this study is meant for manufacturing sector and a discrete study on service sector is recommended to get the full image of the proposed research model. Moreover, incorporating ethical leadership as one dimension in the proposed model as a moderating variable could possibly alter the effect size because ethical leaders are key to strong moral capital [123] Their actions and decisions set the tone, fostering trust and ethical conduct [124]. This, in turn, leads to a more productive and successful organization.

Moreover, the study describes diversity in general terms. It would be valuable for future studies to delve deeper into specific dimensions of diversity (e.g., gender, age, and ethnicity) and assess their unique impacts on organizational performance and workplace ethics.

## Supporting information

**S1 Appendix.**
(DOCX)

## Acknowledgments

We are indebted to the data enumerators who worked relentlessly to acquire targeted data from food and beverage firms. Besides, our appreciation goes to the human resource managers of each firm who helped our data enumerators have a smooth link with respondents; they have assisted us in identifying viable respondents by providing their name list along with ID numbers.

## Author Contributions

**Formal analysis:** Zerihun Ayenew Birbirsa, Gemechu Nemera Dinber.

**Investigation:** Abel Tewolde Mehari.

**Methodology:** Zerihun Ayenew Birbirsa.

**Resources:** Gemechu Nemera Dinber.

**Writing – review & editing:** Gemechu Nemera Dinber.

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
