## [Decision Letter · Decision Letter 0]

30 Nov 2023

PONE-D-23-34803THE EFFECT OF WORKFORCE DIVERSITY ON ORGANIZATIONAL PERFORMANCE WITH THE MEDIATION ROLE OF WORKPLACE ETHICS: EMPIRICAL EVIDENCE FROM FOOD AND BEVERAGE INDUSTRY.PLOS ONE

Dear Dr. Mehari,

Thank you for submitting your manuscript to PLOS ONE. After careful consideration, we feel that it has merit but does not fully meet PLOS ONE’s publication criteria as it currently stands. Therefore, we invite you to submit a revised version of the manuscript that addresses the points raised during the review process.

The research topic is very interesting, but the manuscript needs major revisions.  I cannot guarantee that the revised manuscript will definitely be published.  I hope the constructive feedback provided by the two authors can help improve the manuscript.

We look forward to receiving your revised manuscript.

Kind regards,

Chunyu Zhang

Academic Editor

PLOS ONE

Journal Requirements:

2. You indicated that ethical approval was not necessary for your study. We understand that the framework for ethical oversight requirements for studies of this type may differ depending on the setting and we would appreciate some further clarification regarding your research. Could you please provide further details on why your study is exempt from the need for approval and confirmation from your institutional review board or research ethics committee (e.g., in the form of a letter or email correspondence) that ethics review was not necessary for this study? Please include a copy of the correspondence as an ""Other"" file.

"NO authors have competing interests"

6. Please amend the manuscript submission data (via Edit Submission) to include authors Dr. Zerihun Ayenew and Dr. Gemechu Nemera.

7. Please clarify the number of Tables uploaded in your PDF file. 

Reviewers' comments:

Reviewer's Responses to Questions

**Comments to the Author**

1. Is the manuscript technically sound, and do the data support the conclusions?

Reviewer #1: No

Reviewer #2: Partly

2. Has the statistical analysis been performed appropriately and rigorously? 

Reviewer #1: No

Reviewer #2: Yes

3. Have the authors made all data underlying the findings in their manuscript fully available?

Reviewer #1: Yes

Reviewer #2: Yes

4. Is the manuscript presented in an intelligible fashion and written in standard English?

Reviewer #1: No

Reviewer #2: No

5. Review Comments to the Author

Reviewer #1: Thank you for entrusting me with the review of the manuscript titled ' The effect of workforce diversity on organizational performance with the mediation role of workplace ethics: empirical evidence from food and beverage industry '. While the topic is significant and timely, I have identified several areas where improvements are needed for clarity and coherence:

1. Abstract and Focus: The abstract should concentrate on findings that directly align with your research objectives, namely the impact of workforce diversity on performance and the mediating role of workplace ethics. Therefore, the discussion of these variable relationships is crucial. Devoting extensive space to the results of Confirmatory Factor Analysis (CFA) in the abstract detracts from these essential elements.

2. Keywords Selection: The current keywords do not adequately capture the essence of your research. A thoughtful reconsideration to select keywords that more accurately reflect your study’s focus is recommended.

3. Introduction Coherence: The introduction appears disjointed, with insufficient linkage between paragraphs. Each paragraph is too brief and scattered, lacking a cohesive narrative. Importantly, a clear delineation of the research's contribution and the knowledge gap it addresses is absent.

4. Clarification on EEO and Diversity: The relationship between Equal Employment Opportunity (EEO) and diversity needs to be more explicitly articulated. As it stands, the connection of this discussion to your main theme is unclear.

5. Methodology - Measurement of Diversity: In the methodology section, particularly regarding the measurement of diversity, clarify what PD & RD represent. Also, ensure the three numerical values mentioned are accurately represented and their significance is clear.

6. Acronym Usage: Please provide the full name for OWEI-SF upon its first occurrence in the text to enhance clarity.

7. Clarification in Table 1: The meanings of 'nf' and 'ne' in Table 1 are unclear. An explanation is necessary for reader comprehension.

8. Citation for B-S Bootstrapping: It is imperative to include a citation supporting the use of B-S bootstrapping to correct data non-normality.

9. Structural Reorganization: Considerable reorganization of the manuscript is advisable. Large sections on EFA and CFA tests, which have limited direct relevance to your study, could be better placed in an appendix, with a focus on demonstrating the quality of your measurements.

10. Relevance of SEM Comparisons: The rationale behind comparing models with and without control variables in Structural Equation Modeling (SEM), including different factor structures in CFA, does not seem directly linked to your research objectives.

In conclusion, while the manuscript addresses an important topic, significant improvements in language expression, structural coherence, and alignment of content with research objectives are necessary for a more impactful academic contribution.

Reviewer #2: Introduction and Literature review sections are weak and need rigorous work. For instance, introduction needs to be crafted in way that the problem exists in a society and research gap is there, based on the gap, identify the objectives of the study. .

As far as Literature is concerned, try to explain the variables in greater detail.

Why there is no support of theory in terms of hypotheses development?

In terms of Methods section, the context of Ethiopian beverages should be explained and Stratas must be further explained in detail. Minimum sample test should be conducted as well.

In the end, implications should be explained in detail along with limitations and future directions.

6. PLOS authors have the option to publish the peer review history of their article (what does this mean?). If published, this will include your full peer review and any attached files.

Reviewer #1: **Yes: **I-Hua, Chen

Reviewer #2: No

---

## [Author Response · Author response to Decision Letter 0]

17 Dec 2023

Thank you for the genuine, fair, and constructive comments forwarded from the Academic Editor, Reviewer 1 and Reviewer2.We openly admitted the pinpointed weaknesses in our manuscript and made a change according to these forwarded comments as per the decision letter. Detail and a one to one changes we made on the manuscript is illustrated in the rebuttal letter attached in the revised submission. Kindly, consult the rebuttal letter for specific response to the comments.

---

## [Decision Letter · Decision Letter 1]

9 Apr 2024

PONE-D-23-34803R1THE EFFECT OF WORKFORCE DIVERSITY ON ORGANIZATIONAL PERFORMANCE WITH THE MEDIATION ROLE OF WORKPLACE ETHICS: EMPIRICAL EVIDENCE FROM FOOD AND BEVERAGE INDUSTRY.PLOS ONE

Dear Dr. Mehari,

Thank you for submitting your manuscript to PLOS ONE. After careful consideration, we feel that it has merit but does not fully meet PLOS ONE’s publication criteria as it currently stands. Therefore, we invite you to submit a revised version of the manuscript that addresses the points raised during the review process.

Reviewer 3 suggests explaining the relationship between theory and model, which we think is very important.

We look forward to receiving your revised manuscript.

Kind regards,

Chunyu Zhang

Academic Editor

PLOS ONE

Journal Requirements:

Reviewers' comments:

Reviewer's Responses to Questions

**Comments to the Author**

1. If the authors have adequately addressed your comments raised in a previous round of review and you feel that this manuscript is now acceptable for publication, you may indicate that here to bypass the “Comments to the Author” section, enter your conflict of interest statement in the “Confidential to Editor” section, and submit your "Accept" recommendation.

Reviewer #3: (No Response)

Reviewer #4: All comments have been addressed

2. Is the manuscript technically sound, and do the data support the conclusions?

Reviewer #3: Yes

Reviewer #4: Yes

3. Has the statistical analysis been performed appropriately and rigorously? 

Reviewer #3: Yes

Reviewer #4: Yes

4. Have the authors made all data underlying the findings in their manuscript fully available?

Reviewer #3: No

Reviewer #4: Yes

5. Is the manuscript presented in an intelligible fashion and written in standard English?

Reviewer #3: Yes

Reviewer #4: Yes

6. Review Comments to the Author

Reviewer #3: 1. Keywords should be in Alphabetical order.

2. Explain overarching theory and its connection with you research model.

3. Conclusions and Implications must mention theoretical and practical implications as well. Further, Limitations and future directions for research may also increase the worth of manuscript.

4. Cross check all the citations and references. Some of references do not have DOI address therefore download the citations from page of respective journals to include DOI.

5. Attach data collection instrument (Questionnaire).

Reviewer #4: The authors have incorporated the suggested changes, therefore, now the paper could be accepted for the publication.

7. PLOS authors have the option to publish the peer review history of their article (what does this mean?). If published, this will include your full peer review and any attached files.

Reviewer #3: No

Reviewer #4: **Yes: **Rizwan Raheem Ahmed, Ph.D.

---

## [Author Response · Author response to Decision Letter 1]

14 Apr 2024

The manusript is revised based on the comments from the acadmic editor, and the reveiewers.Details about the changes made are illustrated in the newly attached rebutal letter.Moreover,the track changes version of the revised manuscript and the clean copy of it is also attached.Genuinelly, all the comments are addressed in this new and revised version of our manuscript.Thank you for allowing us to revise and submit the manuscript.

---

## [Editor Report · Decision Letter 2]

3 May 2024

PONE-D-23-34803R2THE EFFECT OF WORKFORCE DIVERSITY ON ORGANIZATIONAL PERFORMANCE WITH THE MEDIATION ROLE OF WORKPLACE ETHICS: EMPIRICAL EVIDENCE FROM FOOD AND BEVERAGE INDUSTRY.PLOS ONE

Dear Dr. Mehari,

Thank you for submitting your manuscript to PLOS ONE. After careful consideration, we feel that it has merit but does not fully meet PLOS ONE’s publication criteria as it currently stands. Therefore, we invite you to submit a revised version of the manuscript that addresses the points raised during the review process.

Please provide ethical approval.

We look forward to receiving your revised manuscript.

Kind regards,

Chunyu Zhang

Academic Editor

PLOS ONE
---

## [Author Response · Author response to Decision Letter 2]

7 May 2024

An ethical approval letter is attached in this review session.

---

## [Decision Letter · Decision Letter 3]

26 Jun 2024

PONE-D-23-34803R3THE EFFECT OF WORKFORCE DIVERSITY ON ORGANIZATIONAL PERFORMANCE WITH THE MEDIATION ROLE OF WORKPLACE ETHICS: EMPIRICAL EVIDENCE FROM FOOD AND BEVERAGE INDUSTRY.PLOS ONE

Dear Dr. Mehari,

Thank you for submitting your manuscript to PLOS ONE. After careful consideration, we feel that it has merit but does not fully meet PLOS ONE’s publication criteria as it currently stands. Therefore, we invite you to submit a revised version of the manuscript that addresses the points raised during the review process.

I suggest the author make revisions based on the comments from Reviewer 5 to increase the readability of the paper. At the same time, this study involves humans, and without providing ethical review evidence, this manuscript will not be published.

We look forward to receiving your revised manuscript.

Kind regards,

Chunyu Zhang

Academic Editor

PLOS ONE

Journal Requirements:

Reviewers' comments:

Reviewer's Responses to Questions

**Comments to the Author**

1. If the authors have adequately addressed your comments raised in a previous round of review and you feel that this manuscript is now acceptable for publication, you may indicate that here to bypass the “Comments to the Author” section, enter your conflict of interest statement in the “Confidential to Editor” section, and submit your "Accept" recommendation.

Reviewer #5: All comments have been addressed

2. Is the manuscript technically sound, and do the data support the conclusions?

Reviewer #5: Partly

3. Has the statistical analysis been performed appropriately and rigorously? 

Reviewer #5: Yes

4. Have the authors made all data underlying the findings in their manuscript fully available?

Reviewer #5: Yes

5. Is the manuscript presented in an intelligible fashion and written in standard English?

Reviewer #5: Yes

6. Review Comments to the Author

Reviewer #5: While the authors conducted extensive statistical analyses, it would be beneficial to provide more details on the assumptions behind these analyses, particularly regarding the normality of data and the decision to use bootstrapping. Explaining how these assumptions were tested and met will strengthen the reliability of the results.

The manuscript discusses diversity in general terms. It would be valuable to delve deeper into specific dimensions of diversity (e.g., gender, age, ethnicity) and their unique impacts on organizational performance and workplace ethics. This could provide more nuanced insights and practical implications for organizations.

The authors should elaborate on the practical implications of their findings for managers and HR professionals. Specific recommendations on how to effectively manage workforce diversity and enhance workplace ethics in the food and beverage industry would make the study more actionable.

While the study is comprehensive, it would benefit from a dedicated section on its limitations and suggestions for future research. Discussing potential limitations, such as the sample being limited to the Ethiopian food and beverage industry, and proposing areas for future studies can help contextualize the findings and encourage further investigation.

Some sections of the manuscript, particularly the methodology and results, are dense and could benefit from clearer and more concise language. Breaking down complex statistical information into simpler terms and using visual aids (e.g., tables, figures) more effectively can enhance readability.

The literature review is thorough, but integrating more recent studies and contrasting them with the current findings could provide a richer context. Highlighting gaps in the existing literature that this study addresses and discussing how the findings contribute to advancing knowledge in the field would be beneficial.

7. PLOS authors have the option to publish the peer review history of their article (what does this mean?). If published, this will include your full peer review and any attached files.

Reviewer #5: **Yes: **Amro Alzghoul

---

## [Author Response · Author response to Decision Letter 3]

29 Jun 2024

Ethical approval letter has been attached on "other" section of submission system on May 07,2024,kindly check this. The comments forwarded from reviewer 5 is addressed by this submission and all required files including clean copy, track changes, rebuttal letter and cover letter files are attached as new version in this session.

---

## [Editor Report · Decision Letter 4]

2 Jul 2024

THE EFFECT OF WORKFORCE DIVERSITY ON ORGANIZATIONAL PERFORMANCE WITH THE MEDIATION ROLE OF WORKPLACE ETHICS: EMPIRICAL EVIDENCE FROM FOOD AND BEVERAGE INDUSTRY.

PONE-D-23-34803R4

Dear Dr. Mehari,

We’re pleased to inform you that your manuscript has been judged scientifically suitable for publication and will be formally accepted for publication once it meets all outstanding technical requirements.

Kind regards,

Chunyu Zhang

Academic Editor

PLOS ONE
---

## [Editor Report · Acceptance letter]

8 Jul 2024

PONE-D-23-34803R4 

PLOS ONE

Dear Dr. Mehari, 

I'm pleased to inform you that your manuscript has been deemed suitable for publication in PLOS ONE. Congratulations! Your manuscript is now being handed over to our production team.

Kind regards, 

on behalf of

Dr. Chunyu Zhang 

Academic Editor

PLOS ONE